# Mortality Related to Chronic Obstructive Pulmonary Disease during the COVID-19 Pandemic: An Analysis of Multiple Causes of Death through Different Epidemic Waves in Veneto, Italy

**DOI:** 10.3390/ijerph191912844

**Published:** 2022-10-07

**Authors:** Ugo Fedeli, Claudio Barbiellini Amidei, Alessandro Marcon, Veronica Casotto, Francesco Grippo, Enrico Grande, Thomas Gaisl, Stefano Barco

**Affiliations:** 1Epidemiological Department, Azienda Zero, Veneto Region, 35131 Padova, Italy; 2Unit of Epidemiology and Medical Statistics, Department of Diagnostics and Public Health, University of Verona, 37134 Verona, Italy; 3Integrated System for Health, Social Assistance and Welfare, Italian National Institute of Statistics, 00184 Roma, Italy; 4Department of Pulmonology, University Hospital Zurich, 8091 Zurich, Switzerland; 5Department of Angiology, University Hospital Zurich, 8091 Zurich, Switzerland; 6Center for Thrombosis and Hemostasis, Johannes Gutenberg University Mainz, 55131 Mainz, Germany

**Keywords:** chronic obstructive pulmonary disease, COVID-19, mortality, multiple causes of death

## Abstract

Mortality related to chronic obstructive pulmonary disease (COPD) during the COVID-19 pandemic is possibly underestimated by sparse available data. The study aimed to assess the impact of the pandemic on COPD-related mortality by means of time series analyses of causes of death data. We analyzed the death certificates of residents in Veneto (Italy) aged ≥40 years from 2008 to 2020. The age-standardized rates were computed for COPD as the underlying cause of death (UCOD) and as any mention in death certificates (multiple cause of death—MCOD). The annual percent change (APC) in the rates was estimated for the pre-pandemic period. Excess COPD-related mortality in 2020 was estimated by means of Seasonal Autoregressive Integrated Moving Average models. Overall, COPD was mentioned in 7.2% (43,780) of all deaths. From 2008 to 2019, the APC for COPD-related mortality was −4.9% (95% CI −5.5%, −4.2%) in men and −3.1% in women (95% CI −3.8%, −2.5%). In 2020 compared to the 2018–2019 average, the number of deaths from COPD (UCOD) declined by 8%, while COPD-related deaths (MCOD) increased by 14% (95% CI 10–18%), with peaks corresponding to the COVID-19 epidemic waves. Time series analyses confirmed that in 2020, COPD-related mortality increased by 16%. Patients with COPD experienced significant excess mortality during the first year of the pandemic. The decline in COPD mortality as the UCOD is explained by COVID-19 acting as a competing cause, highlighting how an MCOD approach is needed.

## 1. Introduction

Data available on the impact of the coronavirus disease 2019 (COVID-19) pandemic on mortality from chronic obstructive pulmonary disease (COPD) are sparse and contradictory. Before the pandemic, mortality from COPD was steadily declining among males in the United States and Europe, whereas limited or no variation was observed among females [1,2,3]. In spite of mortality rates decreasing globally, COPD still represented the third leading cause of death worldwide [4,5]. However, standard mortality statistics based on the underlying cause of death (UCOD) largely underestimate COPD-related mortality due to two main mechanisms: COPD is underdiagnosed and underreported on death certificates [6]; when reported, COPD is rarely listed as the UCOD [7]. The so-called multiple cause of death (MCOD) approach partially overcomes such a limit by analyzing all of the conditions mentioned in the certificate irrespective of their selection as the UCOD. Historically, MCOD data on COPD have been reported in a limited number of countries, including the US [8,9], Brazil [10], England and Wales [11,12], France [13], Spain [14], Italy [7], China [15], and Australia [16].

MCOD-based analyses are warranted to assess the impact of the pandemic on COPD-related mortality, as COVID-19 represents a competing cause of death in statistics focused on the UCOD. Unfavorable COVID-19-related outcomes have been consistently reported in patients with COPD, including increased risks of hospitalization, intensive care unit admission, and death [17]. Despite barriers to healthcare access during the pandemic, a reduced incidence of acute exacerbations of COPD due to the reduced transmission of other respiratory pathogens has been observed [18]. In countries with very limited diffusion of COVID-19 through 2020, mortality from COPD dropped [19], while in countries that experienced huge consequences of the pandemic, deaths from COPD assessed as the UCOD did not change substantially or even slightly declined in 2020, along with the emergency of COVID-19 as a leading cause of death behind heart disease and cancer [18,20,21,22].

Only an MCOD approach allows an assessment of the impact of the COVID-19 epidemic waves on mortality related to underlying chronic conditions at the population level. To our knowledge, such analyses have not been conducted yet for COPD. A proper estimate of COPD-related mortality during the pandemic’s first year is warranted to better understand the impact of subsequent COVID-19 vaccination campaigns on patients with COPD and to contribute, by providing evidence, to favor a rapid public health response and an improved management of future pandemic scenarios. For this purpose, we investigated mortality records in the Veneto region (Northeastern Italy, about 4.9 million inhabitants), one of the first and hardest hit areas by the pandemic [23]. 

## 2. Materials and Methods

This is a population-based study on mortality records examined by means of time series analyses. The mortality register of the Veneto region includes all diseases mentioned in death certificates coded according to the International Classification of Diseases, 10th Revision (ICD-10). The UCOD is selected from all conditions reported on certificates according to rules set by the World Health Organization. To standardize the UCOD assignment, the Automated Classification of Medical Entities (ACME) software was applied to regional records until 2017 [24]. From 2018, the IRIS software was adopted, as in most European countries. The change in software corresponded to the adoption of slightly different rules for UCOD selection based on the 2016 version of ICD-10 [25].

All the death certificates of residents in Veneto aged ≥ 40 years with any mention of COPD (ICD-10 codes J40–J44, J47) were analyzed from 1 January 2008 to 31 December 2020. ICD-10 codes selection was the same adopted in mortality data reported from the European Union [1] and in a previous study carried out in Veneto in 2008–2012 [7]. The age-standardized mortality rates (direct standardization, 2013 European reference population) and proportional mortality (share out of all deaths) were computed for COPD selected as the UCOD, and for any mention in death certificates (MCOD). Changes in trends over pre-pandemic years were assessed using the Joinpoint software (US National Cancer Institute Surveillance Research Program, version 4.8); the average annual percent change in age-standardized rates (APC) with 95% confidence intervals (CI) was estimated for the period 2008–2019 from linear regression models applied to the logarithm of the age-standardized rates.

Trends in the proportion of common comorbidities and complications among death certificates that mentioned COPD were examined as MCOD and UCOD in four separate study periods: 2008–2012, 2013–2017, and 2018–2019 (period when new rules to determine the UCOD started to be applied), and 2020 (first year of the pandemic). The comorbidities and complications investigated were: hypertensive diseases (ICD-10 codes I10–I13), ischemic heart diseases (I20–I25), cerebrovascular diseases (I60–I69), neoplasms (C00–D48), diabetes (E10–E14), dementia/Alzheimer’s disease (F01–F03, G30), COVID-19 (U07.1, U07.2), and common infectious diseases (A00–B99, J00–J22, J69, J85–J86, N10–N12, N13.6, N15.1, N39.0, N70–N76).

The impact of the pandemic on COPD-related mortality was assessed using two approaches. First, the monthly number of COPD-related deaths (based on UCOD and MCOD) observed in 2020 was compared with the corresponding monthly average of the previous two years, when the IRIS software was already adopted for UCOD selection. The ratio of observed to expected deaths (2018–2019 average) was computed with 95% CI based on the Poisson distribution for the entire year and separately for the first (March–May) and second (October–December) epidemic waves experienced in the region by 2020. 

Thereafter, the COPD-related mortality observed in 2020 (MCOD approach) was compared to expected values obtained from the 2008–2019 data. For this purpose, a Seasonal Autoregressive Integrated Moving Average model (SARIMA) was applied to monthly age-standardized mortality rates from 2008 to 2019 and adopted to the forecast figures for the year 2020 by sex and overall. These models accounted for both time trends and the strong seasonality of COPD mortality, which were previously reported [7]. The best fitting SARIMA models were selected on the basis of the lowest corrected Akaike Information Criterion values. The ratio between the observed rates and those predicted by the model was used to estimate excess mortality [26]. The data were analyzed using STATA software (Release 16) (StataCorp, College Station, TX, US).

## 3. Results

Overall, COPD was mentioned in 43,780 death certificates from 2008 to 2020, corresponding to 7.2% of all of the registered deaths among residents aged ≥ 40 years. The proportional mortality for COPD as MCOD declined from 8.2% in 2008 to 6.3% in 2019, and it remained almost unchanged in 2020 (6.2%). In contrast, COPD as the UCOD did not show major changes in proportional mortality from 2008 to 2019 but decreased from 2.6% in 2019 to 2.1% in 2020 (Table 1).

Age-standardized mortality rates related to COPD (MCOD analysis) declined throughout the pre-pandemic period, without changes in trend from 2008 to 2019, as also confirmed by the joinpoint analysis. This decrease was more pronounced among men (APC −4.9%, 95% CI −5.5%, −4.2%) compared to women (APC −3.1%, 95% CI −3.8%, −2.5%). In 2020, an increase was observed in COPD-related mortality, similar in both sexes (Figure 1). In the UCOD analysis, a decrease in mortality was observed up to 2016; thereafter, the mortality rates remained substantially unvaried and decreased again in 2020. This was confirmed among women by the joinpoint analysis that identified two distinct pre-pandemic trends, with rates declining in 2008–2016 (APC −2.3%, 95% CI −3.8%, −0.8%), followed by a non-significant increase in 2016–2019 (APC +3.0%; 95% CI −4.0%, +10.5%). A similar overall pattern, but with a non-significant change in trend, was present among men. 

The reasons for the diverging behavior of the MCOD and UCOD trends can be explained by the data shown in Table 2, which report, only among deaths that mention COPD, the proportion where the disease itself was selected as the UCOD (first row), and the proportion of mention (MCOD) and selection as the UCOD of the most common comorbidities. COPD was selected as the UCOD in 34% of death certificates reporting the disease in 2008–2012 and in 35% in 2013–2017; the increase to 42% in 2018–2019 corresponded to the adoption of new software applying updated selection rules. In 2020, a drop to 34% was due to COVID-19 acting as a strong competing cause for the UCOD, selected in 13% of all death certificates with a mention of COPD. It must be noted that among all deaths registered in the region with COVID-19 as the UCOD (*n* = 5965), COPD was reported in the certificate as a comorbidity in 7.3% of cases (*n* = 441, data not shown). Among comorbidities of COPD, a persistent reduction in ischemic heart diseases was observed from 2008–2012 to 2018–2019 (UCOD: from 16% to 10%; MCOD: from 30% to 23%) as well as for cerebrovascular diseases (MCOD: from 15% to 11%).

The number of deaths from COPD as the UCOD declined by 8% in 2020 compared to the 2018–2019 average; a minor peak in the monthly number of deaths from COPD was observed in April, corresponding to the first epidemic wave (Figure 2). MCOD analyses showed a greater impact of the pandemic: the number of COPD-related deaths in 2020 increased by 14% (95% CI 10–18%). This increase was related to excess mortality during the first epidemic wave in March-May (+20%, 95% CI 12–28%) and to an even larger excess during the second wave in October–December (+49%, 95% CI 41–58%).

The time series analyses based on MCOD confirmed the increase in COPD-related mortality (Appendix A): with respect to those predicted by the SARIMA model, the overall standardized mortality rate observed in 2020 was 16.0% higher. Excess mortality was larger in males (16.8%) than in females (12.3%).

## 4. Discussion

We analyzed the mortality data from the Veneto region in Northern Italy to investigate the impact of the COVID-19 pandemic on COPD-related mortality. The long-term downward trend in mortality, as assessed by the MCOD analysis, was radically modified. In 2020 there was an abrupt increase in mortality compared to the previous years, which was not reflected by figures based on the UCOD. The analysis of monthly mortality rates confirmed the excess of COPD-related mortality during the first and the second epidemic waves. The study confirms that standard mortality statistics relying on the UCOD can result in biased estimates of long-term trends due to changes in coding rules over time and do not allow to properly account for the actual excess mortality during the pandemic due to COVID-19 acting as a competing cause. 

The present study confirms the presence of a long-term pre-pandemic decline in mortality from COPD, with a more favorable trend for men. A 2.56% annual decrease in mortality from COPD between 1994 and 2010 was registered among men in the European Union. This decline was limited to 0.76% in women, with rates even increasing or remaining unchanged in many countries. Consequently, the historical gender gap in COPD mortality tended to decrease, possibly due to diverging trends in smoking habits [1] and to a greater susceptibility in women to secondhand smoke and non-smoking-related factors [3]. Previous peaks in COPD-related mortality were registered in winter seasons characterized by a higher circulation of influenza viruses [27] and severe cold spells [28], as observed in the Veneto region in 2012 within an overall favorable epidemiological scenario (Figure 1). However, based on MCOD analyses, the first year of the COVID-19 pandemic was characterized by an unprecedented rise in COPD-related mortality.

The MCOD analysis also allowed some insights into the changing comorbidity patterns. From 2008–2012 to 2013–2017, the proportion of death certificates with COPD reporting diabetes or neoplasms remained unchanged, whereas those reporting ischemic heart or cerebrovascular diseases declined (Table 2). A reduction over time in the proportion of cardiovascular deaths among patients with COPD has already been reported in England [29]; it must be noted that tobacco is a risk factor for both COPD and cardiovascular diseases, and trends in mortality related to COPD also depend on prevention and appropriate treatment of comorbidities [2].

To date, only sparse data on mortality from COPD during the pandemic’s first year are available, and these are exclusively based on analyses of the UCOD. In the US, the age-standardized mortality rates from chronic lower respiratory diseases (COPD + asthma) decreased by 4.7% in 2020 compared to 2019. However, this finding was heavily influenced by ethnicity, with a larger decline among non-Hispanic Whites than among Hispanics, whereas in non-Hispanic Blacks, rates increased [21]. In Spain, the number of deaths from COPD has decreased from 12,815 in 2019 to 11,786 in 2020, with a corresponding drop in proportional mortality from 3.1% to 2.4%. This finding was due to COVID-19 displacing distribution and the ranking of the leading underlying causes of death [22]. In Scotland and Wales, mortality from COPD through 2020 has continued the decreasing trend observed before the pandemic [18]. In Norway, few deaths due to COVID-19 were registered during the first pandemic wave; meanwhile, deaths from COPD consistently dropped, probably due to the reduction in respiratory infections linked with social distancing, the use of personal protective equipment, and improved hand hygiene [30]. In the preliminary data referring to the first epidemic wave in Italy, the increase in COPD mortality was limited to the most affected region (Lombardy), possibly associated with undiagnosed COVID-19 cases in an overwhelmed health system at the beginning of the pandemic [31].

In the Veneto region, the excess of COPD-related mortality observed in 2020 by means of the MCOD approach was between 14% (when 2018–2019 was used as a reference) and 16% (when compared to the 2008–2019 time series). After a reduction in observed deaths in January–February, probably due to low flu activity, peaks in the number of deaths with a mention of COPD strictly mirrored the COVID-19 epidemic waves affecting the region, the first in spring 2020 and a much larger one starting in autumn 2020. The same time pattern has already been reported for mortality from circulatory and neurological diseases [26,32]. By adopting a similar methodology, the excess mortality estimated for other chronic conditions, such as Parkinson’s disease, in Veneto during 2020 was even larger [26]. Despite the potential beneficial effect of a limited circulation of other respiratory pathogens, the present data demonstrate that patients with COPD represent a population suffering from increased mortality during the COVID-19 epidemic waves. This is also supported by the consistent proportion (above 7%) of the deaths attributed to COVID-19 with COPD mentioned in the certificate. When both diseases are reported as concurrent causes of death, COVID-19 is preferred as the underlying cause over COPD. In fact, UCOD monthly mortality for COPD across 2020 showed only a slight peak corresponding to the first pandemic wave, and no peaks could be observed in the following months. Underreporting of COVID-19 is likely to be responsible for this small peak; during the second pandemic wave, with a complete reporting of COVID-19, no increase in COPD as the UCOD could be observed.

It must be noted that patients with COPD are often affected by other comorbidities that cause them to be frailer. In fact, a recent study suggested that the increase in mortality could be an indirect effect mainly mediated by the presence of concomitant diseases in COPD patients rather than due to COPD itself [33]. Nevertheless, evidence suggests that patients with COPD are at higher risk of developing more severe forms of COVID-19 as well as dying from COVID-19. A recent systematic review reported risks of adverse outcomes such as hospitalizations and admission to the intensive care unit being significantly increased among COPD patients and mortality risks being more than doubled than those of non-COPD patients [34]. 

The main study limitation is represented by the accuracy of death certificates, which are known to be affected by a large underreporting of COPD. Population-based cohort studies with long-term mortality follow-up of individuals with a spirometry-based diagnosis of COPD showed how only a minority of decedents had any mention of the disease in their death certificate, especially in the less severe stages of COPD [6,35]. Furthermore, it is estimated that in adults aged 40 years or older with evidence of persistent airflow limitation on spirometry, only 20–30% have been diagnosed with COPD [36]. The MCOD approach can only partially address the issue of underreporting COPD; however, it is robust to changes in the coding rules and more suitable for assessing time trends [26]. In fact, with the adoption of the IRIS software, the selection of COPD as the UCOD has increased. COPD is more frequently regarded as an obvious cause of infectious diseases (e.g., sepsis, pneumonia), whereas it is no longer considered a consequence of specific cardiovascular diseases, resulting in an overall 19% increase in COPD as the UCOD when using the new classification system [25]. In addition, the impact of COVID-19 as a cause of death in 2020 added further intricacies to the interpretation of standard mortality statistics. Conversely, the MCOD analysis allowed for a complete assessment of the pandemic’s impact on pre-existing long-term mortality trends. 

## 5. Conclusions

During the pandemic’s first year, the age-standardized mortality rates related to COPD increased by 16%, reverting the downward trend observed during 2008–2019. In contrast, the number of deaths from COPD as the underlying cause declined in 2020 due to COVID-19 acting as a competing cause. Only multiple causes of death data allowed us to measure to which extent the pandemic waves in 2020, in the absence of vaccination, affected COPD-related mortality. The results of the present study represent a necessary step for a more accurate estimate of the subsequent positive impact of COVID-19 vaccination on COPD patients. To this purpose, future analyses extended to 2021–2022 data are warranted. 

## Figures and Tables

**Figure 1 ijerph-19-12844-f001:**
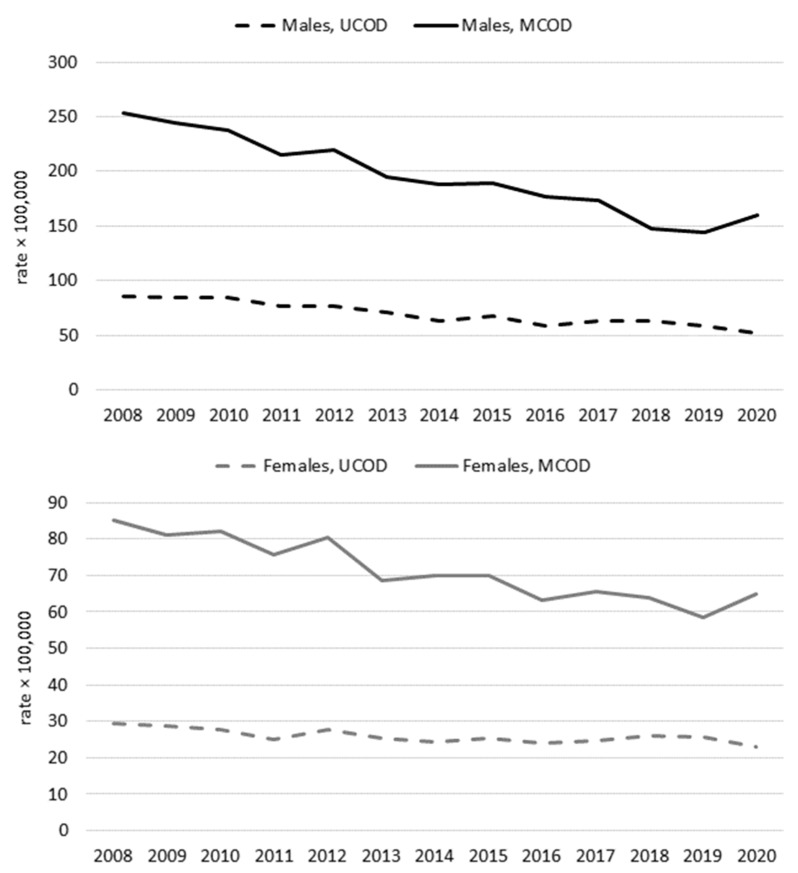
Age-standardized mortality rates (European standard population, age ≥ 40 years) for COPD as the underlying cause of death (UCOD) or mentioned anywhere in the death certificates (MCOD). Veneto region (Italy), 2008–2020. Upper panel males, lower panel females.

**Figure 2 ijerph-19-12844-f002:**
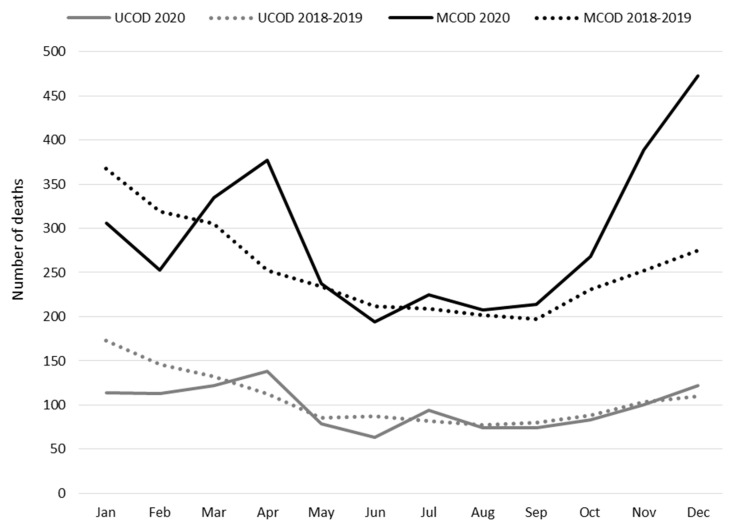
Monthly number of deaths from COPD as the underlying cause of death (UCOD) or mentioned anywhere in the death certificate (MCOD) in 2020, compared to the 2018–2019 average. Residents aged ≥ 40 years, Veneto region (Italy).

**Table 1 ijerph-19-12844-t001:** Number of deaths from COPD selected as the underlying cause of death (UCOD) or mentioned anywhere in the death certificate (MCOD), and the corresponding share of all deaths (proportional mortality). Residents aged ≥ 40 years, Veneto region (Italy), 2008–2020.

	UCOD		MCOD	
Year	*n*	Proportional Mortality	*n*	Proportional Mortality
2008	1179	2.7%	3538	8.2%
2009	1176	2.7%	3475	8.0%
2010	1190	2.7%	3476	8.0%
2011	1118	2.5%	3340	7.6%
2012	1240	2.7%	3645	7.9%
2013	1168	2.6%	3273	7.3%
2014	1123	2.5%	3347	7.5%
2015	1241	2.6%	3473	7.2%
2016	1142	2.4%	3264	6.9%
2017	1219	2.5%	3361	6.9%
2018	1290	2.7%	3088	6.4%
2019	1265	2.6%	3022	6.3%
2020	1175	2.1%	3478	6.2%
2008–2020	15,526	2.6%	43,780	7.2%

**Table 2 ijerph-19-12844-t002:** COPD and common comorbidities/complications (ICD-10 codes) selected as the underlying cause or death (UCOD), or with any mention in death certificates (MCOD), out of all deaths COPD-related deaths. Residents aged ≥40 years, Veneto region (Italy), 2008–2020.

		2008–2012	2013–2017	2018–2019	2020
COPD (J40–J44, J47)	UCOD MCOD	34% 100%	35% 100%	42% 100%	34% 100%
Ischemic heart diseases (I20–I25)	UCOD	16%	13%	10%	8%
	MCOD	30%	25%	23%	21%
Cerebrovascular diseases (I60–I69)	UCOD	5%	4%	4%	4%
	MCOD	15%	12%	11%	11%
Ipertensive disesases (I10–I13)	UCOD	6%	7%	6%	6%
	MCOD	28%	27%	27%	27%
Diabetes (E10–E14)	UCOD	2%	2%	2%	2%
	MCOD	15%	15%	15%	17%
Neoplasms (C00–D48)	UCOD	14%	15%	14%	12%
	MCOD	23%	22%	22%	21%
Dementia/Alzheimer (F01, F03, G30)	UCOD	3%	3%	3%	3%
	MCOD	9%	9%	10%	10%
Infectious diseases (A00–B99, others *)	UCOD	2%	3%	1%	1%
	MCOD	23%	24%	25%	30%
COVID-19 (U07.1, U07.2)	UCOD	-	-	-	13%
	MCOD	-	-	-	15%

* ICD-10 codes: J00–J22, J69, J85–J86, N10–N12, N13.6, N15.1, N39.0, N70–N76.

## Data Availability

The data presented in this study are available on request from the corresponding author. The data are not publicly available.

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
