# Peer review of "Mortality Related to Chronic Obstructive Pulmonary Disease during the COVID-19 Pandemic: An Analysis of Multiple Causes of Death through Different Epidemic Waves in Veneto, Italy"

_ijerph, 2022, doi:10.3390/ijerph191912844_

Round 1
Reviewer 1 Report
This manuscript presents the results of a study of mortality associated with chronic obstructive pulmonary disease during the COVID -19 pandemic in Veneto, Italy. The topic is relevant and the study design and procedure are very clear. The article has clear language and the aim of the study is clear and interesting. I have only minor suggestions for revision:
Figures - usually the title of the figures appears below the figure and not above it. I recommend the authors to change this.
Figure 1- To make the interpretation of the figures clearer, I would put Figure A and Figure B and then indicate in the legend what each figure refers to (e.g. Figure A refers to the male gender data and Figure B refers to the female gender data). In this way, the presentation of results by gender would be clearer.
Line 140 “COPD was selected as the UCOD in 34% of death certificates 140 reporting the disease in 2008-2017”. In Table 2, 34% was reported for COPD in 2008-12 and 35% in 2013-2017, but the authors wrote 34%, so they need to clarify this aspect.
Author Response
Comment 1. Figures - usually the title of the figures appears below the figure and not above it. I recommend the authors to change this.
Answer 1. In the revised manuscript the titles appear below the figures.
Comment 2. Figure 1- To make the interpretation of the figures clearer, I would put Figure A and Figure B and then indicate in the legend what each figure refers to (e.g. Figure A refers to the male gender data and Figure B refers to the female gender data). In this way, the presentation of results by gender would be clearer.
Answer 2. To make presentation of Figure 1 clearer, we changed the legends that now appear “Males, UCOD / Males, MCOD” in the upper panel and “Females, UCOD / Females, MCOD” in the lower panel. Also the title has been modified, specifying that the upper panel refers to males and the lower panel to females
Comment 3. Line 140 “COPD was selected as the UCOD in 34% of death certificates reporting the disease in 2008-2017”. In Table 2, 34% was reported for COPD in 2008-12 and 35% in 2013-2017, but the authors wrote 34%, so they need to clarify this aspect.
Answer 3. The sentence has been modified as follows “COPD was selected as the UCOD in 34% of death certificates reporting the disease in 2008-2012 and in 35% in 2013-2017; the increase to 42% in 2018-2019 corresponded to ….”
Reviewer 2 Report
This manuscript pretend to investigate the impact of the COVID-19 pandemic on COPD-related mortality. It confirms that standard mortality statistics relying on the UCOD can result in biased estimates of long-term trends due to changes in coding rules over time, and do not allow to properly accounting for the actual excess mortality during the pandemic due to COVID-19 acting as a competing cause.
This study main contribution is the usefulness of analysis and description of possible trends in health indicators, like mortality, with different ways of measures (UCOD; MCOD) per temporal waves, which can produce biases and interfere with the monitoring of health policies and in turn with healthcare decision-making. A weakness aspect is the uncertainty of the differences in the classification of diseases and in the calculation of indicators or divergences in geographic areas (when compare with another countries) which may make it impossible to compare data.
In general, the manuscript is clear and relevant for the field of public health because present contradictory data of mortality rates of COPD during the COVID-19 pandemic and the authors have reported these facts in an adequate manner. They well stablish the different methods used, like the registers of residents aged >= 40 years in Venetto region, the international classification of diseases and the UCOD rules according WHO and highlight how a MCOD approach is needed. The software adopted in most European countries was the IRIS, but in this case, the authors need to explain better it´s specifications to enable the reproducibility of the results.
There are specifically aspects in which the manuscript can be improved:
Abstract: need to define more clearly the objectives of the study and the methods, clarifying the study design.
Introduction: The problem was well identified and but it was not clear enough the importance of this study. Why it is important nowadays to know about the impact of COVID-19 in mortality of COPD and other comorbidities?
Materials and methods: specify the study design
Results: the outcomes was reported by period (2008-2019 and 2020), by sex in the same periods. The rates declines in 2008-2016, followed by a non-significant increase in 2016-2019 among men and women. Then, the authors report the mortality rates for comorbidities in the periods from 2008-2017; 2018 to 2019 and 2020 (table 2) but it is not clear why COPD thus not has results comparing UCOD and MCOD from 2018 similarly other diseases.
Discussion: here is clear the objective, but the results of previous investigations are a little bit confused in terms of structure. Therefore, we suggest that: first, initiate with a summary of the principal results of your study; second, the interpretation of mortality rates of COPD previous of pandemic and along the periods stablished; third, the mortality rates of comorbidities previous and during COVID-19 pandemic; Fourth, the confirmation of these results in Veneto region and in other countries and measurement methods are the same (comment the biased results); fifth, present a more critical analysis for the different results and explain the factors related to the geographic context of the listed countries (eg. COPD underdiagnosis).
Conclusions: It must be improved considering the field of knowledge of this study in the area of public health. The advantages of vaccination against seasonal flu, covid-19 and other recommended vaccines that have reduced mortality from both diseases must be presented.
References. the cited references are mostly recent publications (within the last 5 years) and relevant and it does not include self-citations.
Author Response
Comment 1. Abstract: need to define more clearly the objectives of the study and the methods, clarifying the study design.
Answer 1. Objective of the study and study design are now clarified in the new second sentence of the Abstract: “The study aim is to assess the impact of the pandemic on COPD-related mortality by means of time series analyses of causes of death data.”
Comment 2. Introduction: The problem was well identified and but it was not clear enough the importance of this study. Why it is important nowadays to know about the impact of COVID-19 in mortality of COPD and other comorbidities?
Answer 2. The last paragraph of the Introduction section has been expanded. It is now specified that a proper estimate of COPD-related mortality during the pandemic’s first year is warranted to assess the impact of COVID-19 vaccination in patients with COPD, and to properly design public health measures in future pandemics.
Comment 3. Materials and methods: specify the study design
Answer 3. The section’s first sentence now states that the study is based on time series analyses of mortality records.
Comment 4. Results: the outcomes was reported by period (2008-2019 and 2020), by sex in the same periods. The rates declines in 2008-2016, followed by a non-significant increase in 2016-2019 among men and women. Then, the authors report the mortality rates for comorbidities in the periods from 2008-2017; 2018 to 2019 and 2020 (table 2) but it is not clear why COPD thus not has results comparing UCOD and MCOD from 2018 similarly other diseases.
Answer 4. Table 2 is now more clearly explained in Results of the revised manuscript. It is now specified that the Table reports, only among deaths with mention of COPD, the proportion where the diseases itself was selected as the UCOD (first row), and the proportion of mention (MCOD) and selection as the UCOD of the most common comorbidities.
Comment 5. Discussion: here is clear the objective, but the results of previous investigations are a little bit confused in terms of structure. Therefore, we suggest that: first, initiate with a summary of the principal results of your study; second, the interpretation of mortality rates of COPD previous of pandemic and along the periods stablished; third, the mortality rates of comorbidities previous and during COVID-19 pandemic; Fourth, the confirmation of these results in Veneto region and in other countries and measurement methods are the same (comment the biased results); fifth, present a more critical analysis for the different results and explain the factors related to the geographic context of the listed countries (eg. COPD underdiagnosis).
Answer 5. We acknowledge that the structure of the Discussion was a bit confused in the original submission. Now the section has been re-arranged, maintaining the original contents, in the following parts: summary of main results; comments on pre-pandemic mortality rates and on pre-pandemic trends in comorbidities, with discussion of the relevant literature; detailed discussion of mortality rates during the pandemic, compared with the available literature; study limits including the main issue of COVID under-diagnosis.
Comment 6. Conclusions: It must be improved considering the field of knowledge of this study in the area of public health. The advantages of vaccination against seasonal flu, covid-19 and other recommended vaccines that have reduced mortality from both diseases must be presented.
Answer 6. The Conclusion has been expanded, specifying that analyzed data can answer to a preliminary fundamental question, “to which extent pandemic waves in 2020, in the absence of vaccination, affected COPD-related mortality?” This is necessary to better measure the positive impact that COVID-19 vaccines had on COPD patients, by means of analyses extended to 2021-2022 data.